**Quasi 10-day wave modulation of equatorial ionization anomaly during the Southern**
**Hemisphere stratospheric warming of 2002**
Xiaohua Mo[1],Donghe Zhang[2*]
[1]College of Science, Key Laboratory for Ionospheric Observation and Simulation, Guangxi University for
Nationalities, Nanning, China
[2]Department of Geophysics, Peking University, Beijing, 100871, China
Correspondence: Donghe Zhang (zhangdh@pku.edu.cn)
**Abstract**
The present paper studies the perturbations in equatorial ionization anomaly (EIA) region during the
Southern Hemisphere (SH) sudden stratospheric warming (SSW) of 2002, using the location of EIA crests
derived from Global Positioning System (GPS) station observations,the Total Electron Content (TEC)
obtained by International GNSS Service (IGS) global ionospheric TEC map (GIMs), and the equatorial
electrojet (EEJ) estimated by geomagnetic field in Asian sector. The results indicate the existence of an
obvious quasi 10-day periodic oscillation in the location and TEC of northern and southern EIA crest. An
eastward phase progression of quasi 10-day wave producing the SH SSW of 2002 is also identified in polar
stratospheric temperature. Previous studies have shown that a strong quasi 10-day planetary wave with
zonal wave numbers s=1 extend from the lower stratosphere to mesosphere and lower thermosphere during
the SH SSW of 2002 (Palo et al., 2005). Moreover, the EEJ driven by equatorial zonal electric field
exhibits quasi 10-day oscillation, suggesting the enhanced quasi-10-day planetary wave associated with
SSW penetrates into the ionosphere E region and produces oscillation in EIA region through modulating
the E-region electric fields. Our results reveal some newer features of ionospheric variation that have not
been reported during Northern Hemisphere (NH) SSWs.



## 1. Introduction

Sudden stratospheric warming (SSW) is large-scale meteorological process in the polar stratosphere which is characterized by rapid rise in temperatures and deceleration/reversal in the zonal mean flows (Scherhag, 1952). The primary driver of SSW is thought to be a rapid growth of quasi-stationary planetary wave interacting with zonal mean flow (Matsuno, 1971). Although the main processes of SSW occur in the polar middle atmosphere, its effects on the ionosphere have been observed in significant changes of equatorial electrojet (EEJ), vertical plasma drift, and equatorial ionization anomaly (EIA) (Vineeth et al., 2007; Chau et al., 2009; Goncharenko et al., 2010; Pancheva and Mukhtarov, 2011; Jin et al., 2012).

Since stationary planetary waves in the Southern Hemisphere (SH) generally have smaller amplitudes than in the Northern Hemisphere (NH) where orographic and thermal forcing is stronger (Andrews et al., 1987), major SSWs often occur in NH. Therefore, most studies about SSW effects on the ionosphere are during NH SSW period. In recent years, a great deal of research has been focused on the variation of the low latitude ionosphere during SSW period in the northern hemisphere, and the quasi-16-day periodic disturbance and the lunar tide characteristics have been found in some ionospheric parameters, for example, EEJ, vertical plasma drifts, ionospheric electron density (Vineeth et al., 2007; Chau et al., 2009; Pedatella and Forbes, 2009; Goncharenko et al., 2010). Some researchers considered that this kind of quasi-16-day periodic variations is related to the enhanced planetary wave during the SSW period (Chau et al., 2009; Pedatella and Forbes, 2009; Liu et al., 2010). But others believe that this kind of quasi-16-day period is related to the semi-diurnal lunar tides (Fejer et al., 2010; Park et al., 2012). The direct evidence is the typical SSW feature appears in some ionospheric parameters as morning enhancement and afternoon decrease, in a semi-diurnal pattern that progresses to later local times within several days. However, some studies believe that the local time variation characteristics are not necessarily caused by the semidiurnal lunar tides, Pedatella et al. (2012) demonstrate that the phase of the semidiurnal solar tide changes in a manner that makes it similar to the phase of the lunar semidiurnal tide. Besides, although many studies on the variation of the low latitude ionosphere during the SSW period, the physical connection between the SSW in the polar region and the featured variations in low latitude ionosphere is still not clear. Some studies even consider that the SSW and the featured variation of low latitude ionosphere is co-source, which is the effect of enhanced planetary waves in different regions (Stening et al., 2011).

In comparison, the atmospheric parameters in the mesosphere and lower thermosphere (MLT) are very limited. The atmospheric variation in MLT is usually indirectly reflected through the EEJ obtained from the

geomagnetiic field data in equatorial and low latitude region. The EEJ is driven by zonal electric field which also produces EIA via upward E×B drift (fountain effect). The zonal electric field modulated by tidal winds in the lower thermosphere through the E-region dynamo process is easy to be influenced by various atmospheric waves, so these ionospheric variations often display similar semidiurnal pattern and 13- to 16-day wave signatures which have been associated with enhanced planetary wave, solar and lunar tide wave during SSW period (Pedatella and Forbes, 2009; Goncharenko et al., 2010; Fejer et al., 2010;Park et al., 2012).

In August to September 2002, three minor SSWs and a major SSW appeared in SH (Varotsos 2002; Baldwin et al., 2003). There is sufficient evidence that a series of unusual atmospheric states occurred in this period, i.e., planetary wave scale quasi 10-day variation (Krüger et al., 2005; Palo et al., 2005), short-term semidiurnal tide variability with zonal wave number s=1 (Chang et al., 2009) and the winds oscillation with ~14-days period (Andrew et al., 2004), are all linked to the extremely large planetary wave events. Although the atmospheric activity in connection with 2002 SH SSW has been well revealed in observations and numerical modeling, relatively little is known about the ionosphere effects of 2002 SH SSW. Recently, Olson et al. (2013) studied the equatorial electrodynamic perturbations in Peruvian sector during 2002 SH SSW and found enhanced quasi 2-day fluctuations and large amplitude multi-day perturbations in EEJ and vertical drifts. The research of ionospheric behavior during SH SSW periods are useful for verifying the existing explanation about the origin of ionospheric perturbations during NH SSW periods and revealing some newer features of ionospheric variation, so further investigation of 2002 SH SSW effect on ionosphere with more ionospheric parameters is still warranted.

In the present study, we present the first observational evidence of quasi 10-day oscillation in EIA region during 2002 SH SSW which has not been reported during NH SSWs, based on the location of EIA crests derived from Global Positioning System (GPS) station observations, the Total Electron Content (TEC) obtained by International GNSS Service (IGS) global ionospheric TEC map (GIMs), and the EEJ estimated by geomagnetic field in Asian sector.

**2. Data and Methods**

The location of EIA crests derived from GPS observations are used to analyze the variation in EIA region during 2002 SH SSW from July 20, 2002 to October 27, 2002. The GPS stations are GUAN (23.19°N, 113.34°E, MLAT~12.52°N) and BAKO (6.49°S, 106.84°E, MLAT~17.18°S) which are near

northern and southern EIA crest, respectively. The locations of the GPS stations are shown in Figures 1.
Since the ionospheric vertical TEC usually reach the maximum at EIA crest, the location of EIA crest can
be obtained by vertical TEC values at each ionospheric penetration point (IPP), which is the intersection of
the line of sight and the ionospheric shell (assumed to be 400 km) (Mo et al., 2014). The relative accuracy
of the TEC is 0.02 total electron content unit (1TECU=$10^{16}$ el m$^{-2}$) (Hofmann-Wellenhof et al., 1992). The
sample rate of these GPS stations were 30s, so the resolution of the location of EIA crest is less than 25 km
(Mo et al., 2017). Figures 2a and 2b show the daily average geomagnetic latitude (MLAT) of northern and
southern EIA crests during 2002 SH SSW.
The TEC from GIMs are also used to analyze the variation in EIA region. The GIMs provides maps of
TEC obtained from a global network of GPS receivers, which have temporal resolution of 2 hours and
spatial resolution of 5° in longitude and 2.5° in latitude (Mannucci et al., 1998). The EIA crest usually
reaches its maximum development near 14:00 LT (Huang et al., 1989; Yeh et al., 2001), so the daily
average TEC obtained by GIMs at 12~14 LT, ±5°~±15° MLAT, 100º~150ºE every day in Asian sector are
used to describe the variation in northern and southern EIA region, the results are shown in Figures 2c and
2d.
To demonstrate the dynamical process in EIA region, the EEJ is also used in this study, which can be
estimated by the difference between the horizontal component of geomagnetic field at TIR (8.7°N, 77.8°E,
MLAT~0.03°N) and VSK (17.68°N, 83.32°E, MLAT~8.56°N) (Rastogi et al., 1990). The results are shown
in Figures 2e. In addition, the polar stratospheric temperature (90°S, 10hPa) and zonal mean zonal winds
(60°S, 10hPa) obtained from National Centers for Environment Prediction (NCEP) are used to examine the
extent of the SSW, the results are shown in Figures 2f and 2g. The background of geomagnetic activity
index (Kp) and solar flux index (F10.7) from the websites http://spidr.ngdc.noaa.gov/ are depicted in
Figures 2h and 2i.
**3. Results and Analysis**
It can be seen from Figures 2f and 2g that there were three obvious minor SH SSW events around day
number 230-260 and a major SH SSW event around day number 263-288 (Olson et al., 2013). Figure 3
shows the contour map of polar stratospheric temperature (80°S, 10hPa) obtained from NCEP from July 20,
2002 to October 27, 2002. An eastward phase progression of quasi 10-day wave is clearly observed around
day number 210-270. With SABER temperature data, Palo et al. (2005) also observed similar disturbance
and suggested it consists of an eastward-propagating quasi 10-day wave with zonal wave numbers s=1
superimposed upon a large stationary planetary wave with s=1.

Now we examine the impact of this quasi 10-day wave on EIA region. 2002 was at solar maximum

phase, the ionosphere maybe exhibit some variations caused by periodic solar irradiance and high speed
solar streams related to solar rotation during 2002 SH SSW event, for example, 27-day periodic variation.
To exclude these long period fluctuations in EIA region, the periods longer than 15 days in the MLAT
location and TEC of EIA crest, and EEJ are removed. Specifically, these parameters are subtracted from
their respective 15-day moving average. The residuals are subjected to Lomb-Scargle (L-S) spectral
analysis (Lomb,1976; Scargle, 1982), and the results are shown in Figures 4a, 4b, 4c, 4d, and 4e. The
horizontal dashed lines represent the 95% confidence level. It is evident that the MLAT location and TEC
of EIA crest, and EEJ all exhibit significant quasi 10-day periodic component, which exceed or approach
95% confidence level, suggesting that the whole dynamical process in EIA region is modulated by quasi
10-day wave. Figures 4f and 4g show the L-S spectral analysis of Kp and F10.7. It can be seen that both Kp
and F10.7 do not exhibit quasi 10-day periodic component, indicating that variation in the solar irradiance
and geomagnetic activity cannot account for this quasi 10-day oscillation in EIA region.

To investigate the time evolution of quasi 10-day periodic variation, the Morlet wavelet spectral

analysis is applied to MLAT location and TEC of EIA crest, EEJ and Kp. The periods longer than 15 days
in the MLAT location and TEC of EIA crest , and EEJ are removed before the wavelet spectra is generated,
and the results are illustrated in Figures 5a, 5b, 5c, 5d, and 5e. The black solid contours in each panel
indicate a significance level higher than 95%. The white line in each panel represents the cone of influence
of the wavelet analysis. The color bar number is the power strength for each parameter. Obviously, the most
predominant periodic component in the MLAT location and TEC of EIA crest, and EEJ are quasi 10-day
period, which mainly appeared around day number 210-290, indicating quasi 10-day oscillations in EIA
region go through three minor SSWs and a major SSW period. The time evolution of the power in MLAT
location and TEC of northern EIA crest match well those of southern EIA crest, respectively. In addition,
we note both the MLAT location and the TEC of EIA crest show the quasi 2-day oscillations during major
SSW period (around day number 260-270), which are also found on equatorial ionospheric electric fields
and currents at the same period (Olson et al., 2013). Figure 5f shows the wavelet spectral analysis of Kp
index. It can be seen that quasi 10-day periodic component is nearly absent in Kp around day number
230-290, suggesting that magnetic activity should not be the driving force for this quasi 10-day oscillation
in EIA region.
In order to demonstrate the phase relationship of the quasi 10-day oscillations between northern and
southern EIA crests, the band-pass filter is performed on the MLAT location and TEC of EIA crest. The
absolute values of the MLAT location of EIA crest are used. The band-pass filter is centered at the period of
10-day, with half-power points at 8-day and 12-day, and the results are shown in Figure 6. The quasi 10-day
wave amplitudes of northern and southern EIA crests are roughly equivalent, which exceed 1.7 degree for
MLAT location and 7 TECU for TEC, respectively. Although the quasi 10-day wave of northern EIA crest
match well those of southern EIA crest, the wave of northern EIA crest seemed to delay behind southern
EIA crest, especially for MLAT location. To further verify this, Figure 7 shows the cross-correlation of
quasi 10-day waves in MLAT location (a) and TEC (b) between northern and southern EIA crests. The
cross-correlation coefficients of MLAT location and TEC reach 0.8 and 0.93, respectively. Moreover, the
maximum cross-correlation coefficients for MLAT location is at 1 day, indicating that the wave of northern
EIA crest delay 1 day behind southern EIA crest. This phase difference between northern and southern EIA
crests may be due to differences in longitude between two GPS stations.

**4. Discussions**
In recent years a series of reports have focused on ionospheric perturbations during SSW event. The
most predominant features in low latitude ionosphere associated with SSW event are semidiurnal pattern
and 13- to 16-day periodic variations, which are attributed to nonlinear interaction of planetary wave, solar
and lunar tide wave (Pedatella and Forbes, 2009;Goncharenko et al., 2010; Fejer et al., 2010;Park et al.,
2012). As major SSW often occurs in NH, most studies about SSW effects on the ionosphere are during
NH SSW period. In August to September 2002, the first major SSW was observed in SH. The NH and SH
SSW occurred in Arctic and Antarctic winter, respectively, so the occurring time and location of SH SSW
are opposite to those of NH SSW. The researches of ionospheric behavior during SH SSW periods are
useful for testing the general rule of ionospheric perturbations during NH SSW periods. For example,
Olson et al. (2013) demonstrated that multi-day ionospheric perturbations responding to 2002 SH SSW
resemble those observed during NH SSWs and these ionospheric perturbations were associated with
enhanced lunar tidal effects.
In this study we present observations of quasi 10-day oscillation in EIA region during the 2002 SH
SSW that has not been reported during NH SSWs. This quasi 10-day periodic component is absent or very
weak in Kp and F10.7 index, indicating that the magnetic activity and solar irradiance cannot account for
this quasi 10-day oscillation in EIA region. Meanwhile, an unusual atmospheric state occurred in this
period that the ozone hole over the Antarctic has a smaller size and splits into two separate holes (Varotsos
2002; Baldwin et al., 2003). This phenomenon is thought to be due to high temperatures in the Antarctic
stratosphere, which was contributed to by upward propagation of planetary waves (Venkat Ratnam et al.,
2004). Moreover, strong planetary wave scale quasi 10-day variation was observed in polar stratospheric
temperature during this period. The wave interactions between eastward-propagating waves with periods
near 10 days, quasi-stationary planetary waves, and the zonal mean atmospheric state were eventually
driven towards total break-down of the polar vortex and a major warming of the stratosphere (Krüger et al.,
2005; Palo et al., 2005). So the quasi 10-day oscillations in EIA region should be ascribed to atmosphere
perturbations linking the SSW in the Southern Hemisphere.

The coupling process of 10-day oscillation between the lower atmosphere and ionosphere can be

demonstrated by existing observations and simulations. A series of studies have showed how the quasi
10-day planetary wave in stratosphere can penetrate into the ionosphere E region (Krüger et al., 2005; Palo
et al., 2005; Chang et al., 2009). Krüger et al. (2005) revealed the eastward-traveling waves with periods
near 10 days and their interaction with quasi-stationary planetary waves forced in the troposphere during
2002 SH SSW event, supporting the observational and numerical evidence that the eastward traveling wave
interacts with the stationary wave to produce a quasi-periodic amplitude modulation of the stationary waves
(Hirota et al., 1990; Ushimaru and Tanaka, 1992). Palo et al. (2005) found an eastward-propagating quasi
10-day wave with zonal wave numbers s=1 and s=2, and a quasi-stationary planetary waves with s=1
extend from the lower stratosphere to the 100-120 km height region with little amplitude attenuation. While
the quasi-stationary planetary wave is confined to the high latitude atmosphere and cannot directly
propagate to equatorial ionosphere, the tides were introduced into planetary wave modulation mechanism.
Eswaraiah et al. (2018) reported that zonal diurnal and semidiurnal tide amplitudes in Antarctica
mesosphere and lower thermosphere were enhanced around day number 230-290 during 2002 SH SSW,
which coincides with the enhanced period of quasi 10-day oscillations in EIA region shown in Figure 5.
Moreover, Chang et al. (2009) showed that the short-term variability of the s=1 semidiurnal tide is strongly
dependent upon the PW1 events (quasi-10-day wave) prior to the major warming during 2002 SH SSW,
supporting the suggestion that the quasi-stationary planetary wave can influence migrating and
nonmigrating solar tides globally (Liu et al., 2010; Pedatella and Forbes, 2010). So the interactions between
quasi-10-day planetary wave and tide will modify the ionosphere E-region winds, which can produce
E-region electric fields via the E-region dynamo process. In this study, the EEJ driven by equatorial zonal
electric field also exhibits quasi 10-day oscillation, indicating that the upward-propagating planetary waves
interacted with the tide produced oscillation in EIA region through modulating E-region electric fields.
Specifically, the E-region electric fields map to lower ionospheric F-region along the magnetic field lines
and generate an eastward electric field (Goncharenko et al., 2010). At the magnetic equator, the eastward
electric field with quasi 10-day periodic variation change electron density distribution in the low-latitude
region via E×B drift, and finally leads to quasi 10-day planetary waves characteristic variations in EIA
region. Previous studies have revealed a strong correlation between ionospheric perturbations and the
occurrence of NH SSW. During NH SSW period, quasi 16-day oscillations and semidiurnal pattern are
observed in equatorial mesopause temperature, the MLT meridional and zonal wind, EEJ, electron density
and TEC (Vineeth et al., 2007; Pedatella and Forbes, 2009; Park et al., 2012 ; Jonah et al., 2014). Some
researchers attribute these ionospheric perturbations to the strong dynamical coupling between the lower
atmosphere and ionosphere through the intensification of planetary wave activity (Chau et al., 2009), lunar
(Fejer et al., 2010) and solar (Pedatella et al., 2012) tide. In this study, the consistent quasi 10-day
oscillations appear in EEJ, the location and TEC of northern and southern EIA crest, indicating that
coupling mechanism between the lower atmosphere and ionosphere during SH SSW period is consistent
with that during NH SSW period.

In our prior studies, a 14- to 15-day wave during several NH SSW events is ascribed to lunar tide

(Mo et al., 2018). So the source of quasi 10-day oscillations in EIA region during 2002 SH SSW is different
from 14- to 15-day waves during NH SSW. For this 10-day periodic event, it seems that the effect of the
planetary wave is more obvious. Moreover, no obvious 14- to 15-day oscillation is found in EIA region
during 2002 SH SSW, which may be that the equatorial lunar semidiurnal effects during
September-October are weaker than that during January-February (Stening et al., 2011; Pedatella, 2014).
Olson et al. (2013) also reported that the perturbations amplitude of EEJ and vertical drifts modulated by
lunar semidiurnal tides during SH SSW are smaller than those during NH SSW.

**5. Conclusions**

Using the location and TEC of EIA crests derived from GPS station observations and GIMs, we found

a quasi 10-day periodic variability in northern and southern EIA region in Asian sector during the SH SSW
of 2002. In the same time period, this quasi 10-day oscillation is also seen in the polar stratospheric

temperature, which is absent and weak in Kp and F10.7 index, respectively. The SH SSW of 2002 itself is generated by quasi 10-day planetary wave. Previous studies have shown that a strong quasi 10-day planetary wave with zonal wave numbers s=1 extend from the lower stratosphere to mesosphere and lower thermosphere during the SH SSW of 2002 (Palo et al., 2005). Moreover, the EEJ driven by equatorial zonal electric field exhibits quasi 10-day oscillation, indicating that the upward-progagating planetary waves interacted with the tide will modify E-region electric fields, thereby altering the plasma structures through upward E×B drift, which results in the periodical variations in these ionospheric parameters in F region. These results support the suggestion that the quasi 10-day variation in EIA region should be ascribed to enhanced 10-day planetary wave in lower atmosphere associated with SSW.

**Acknowledgements:** The GPS data were from the Crustal Movement Observation Network of China (via e-mail to yglyang@cma.gov. cn) and IGS (available at http://sopac. ucsd.edu). The geomagnetic data at TIR and VSK were from WDC for Geomagnetism, Kyoto (available at http://wdc.kugi.kyoto-u.ac.jp/hyplt/index.html). The GIMs were downloaded from the site ftp://cddis.gsfc.nasa.gov. This research was jointly supported by the National Natural Science Foundation of China (41864006, 41674157, and 41464006), Guangxi Natural Science Foundation (2016GXNSFAA380132), and Chinese Meridian Project. We gratefully acknowledge National Center for Environmental Prediction (NCEP) for providing public access to stratospheric data (available at https://www. esrl.noaa.gov/psd/data/reanalysis/).

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

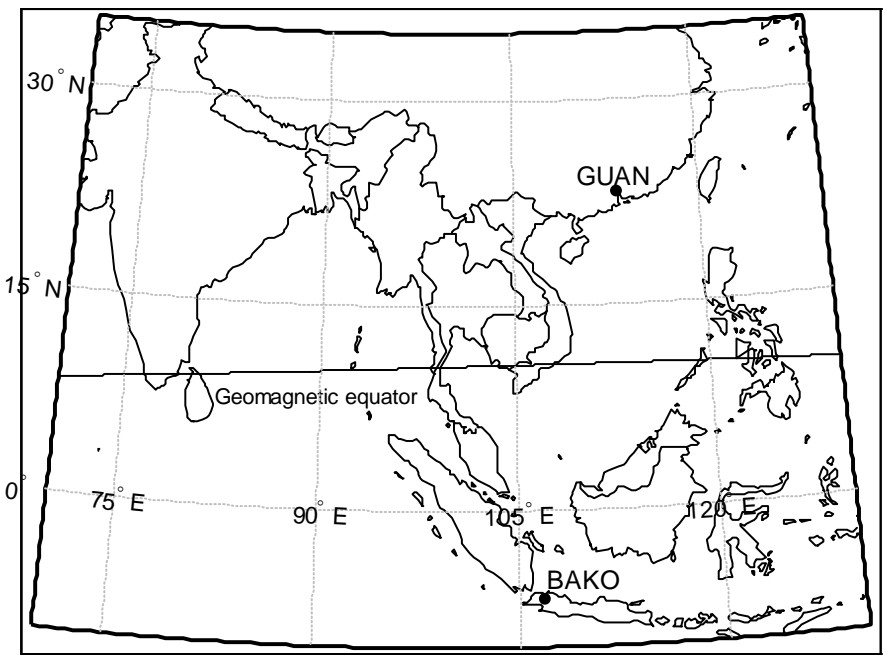


**Figure 1.** Location of the GPS stations in Asian sector.

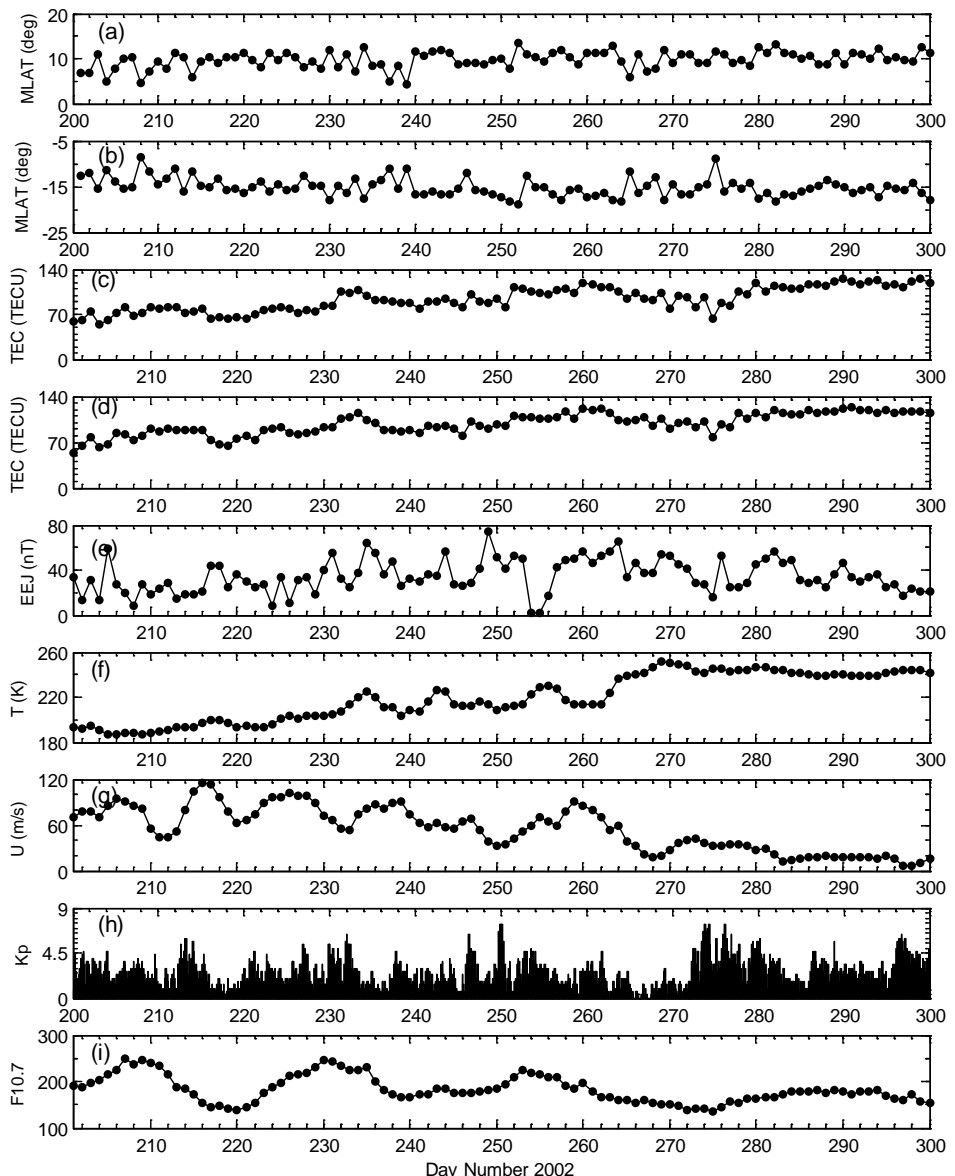


**Figure 2.** The magnetic latitude (MLAT ) location of (a) northern and (b) southern equatorial ionization

anomaly (EIA) crest; The TEC of (c) northern and (d) southern EIA crest; the (e) equatorial electrojet (EEJ),

(f) polar stratospheric temperature (at 90°S, 10hPa) and (g) zonal wind (at 60°S, 10hPa) from National

Centers for Environment Prediction; the (h) Geomagnetic activity index, Kp and (i) solar flux index F10.7

during the period from July 20, 2002 to October 27, 2002.

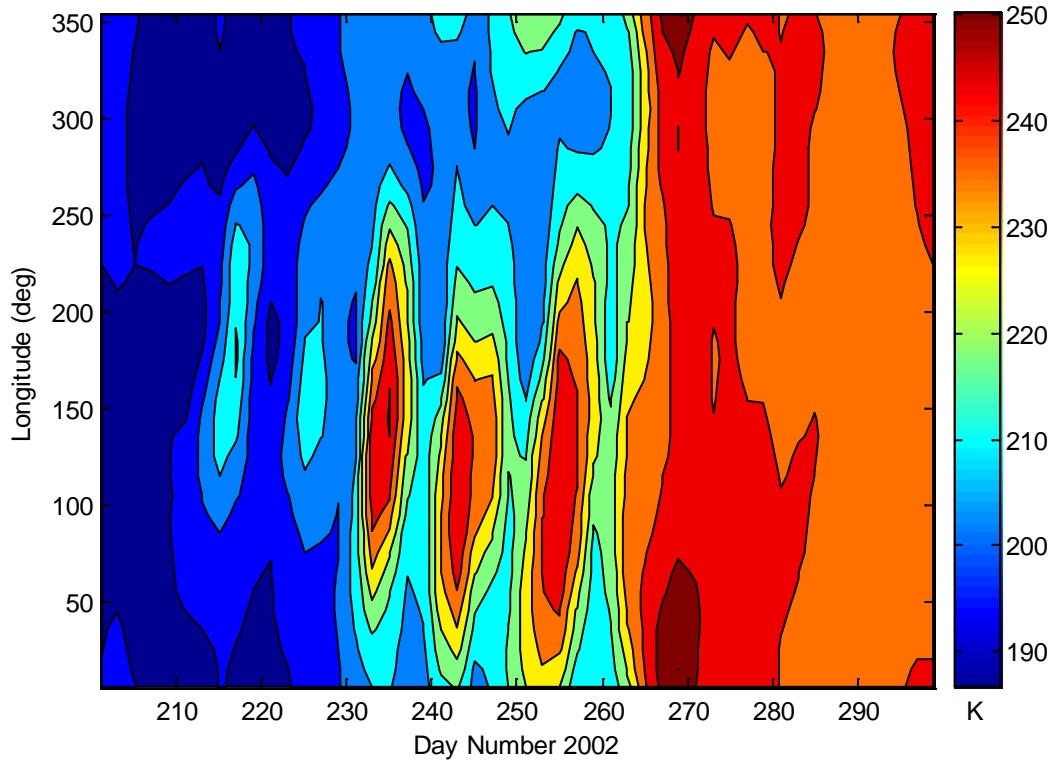


**Figure 3.** The contour map of polar stratospheric temperature (80°S, 10hPa) obtained from NCEP during
the same period as in Figure 2.

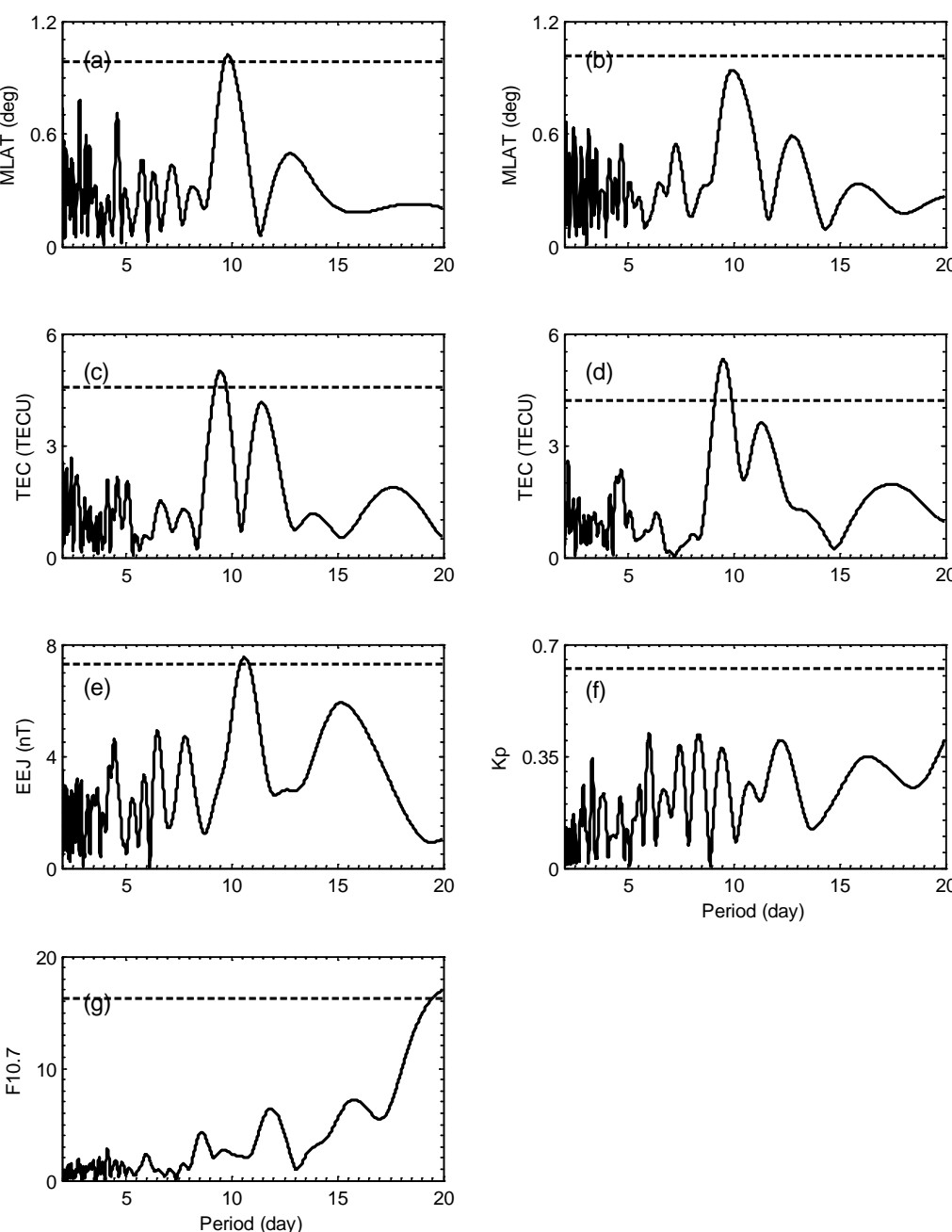


**Figure 4.** Lomb-Scargle periodgrams of the MLAT location of (a) northern and (b) southern EIA crest, the
TEC of (c) northern and (d) southern EIA crest, (e) EEJ, (f) Kp index and (g) F10.7 during the same period
as in Figure 2.

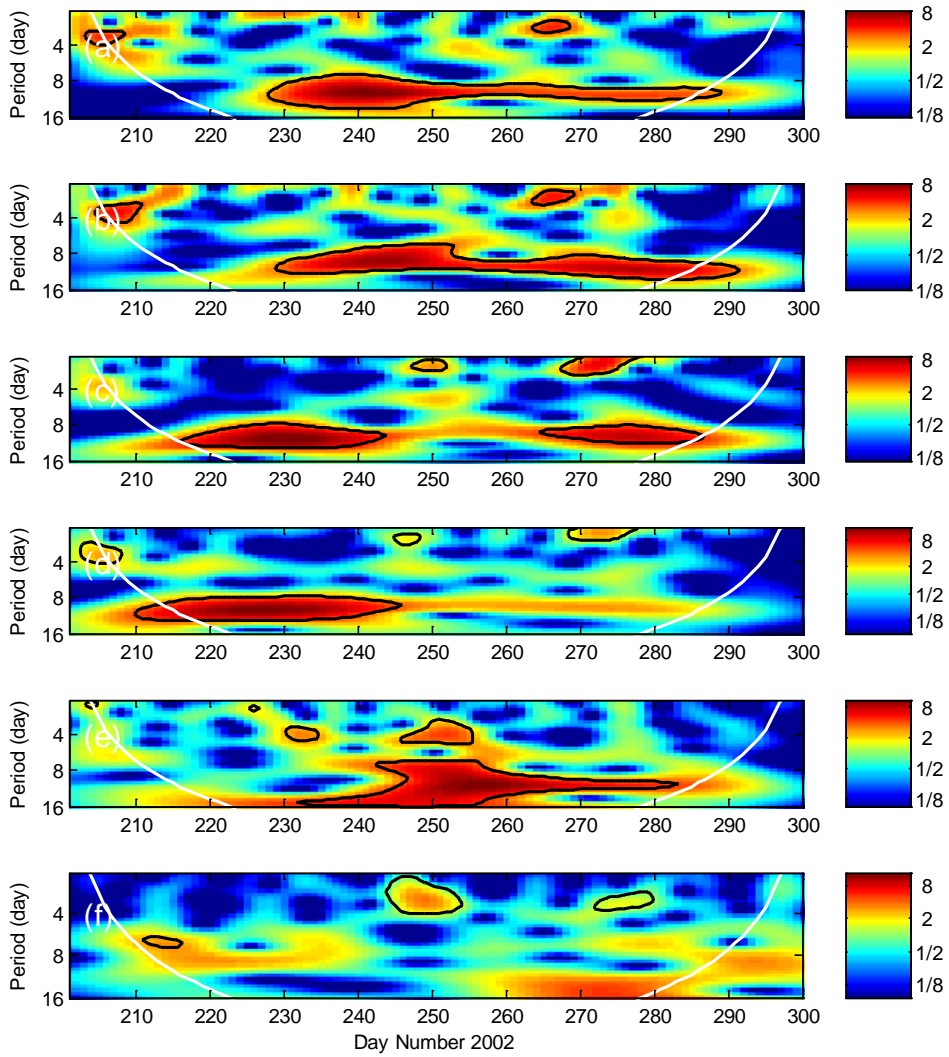


**Figure 5.** The wavelet power spectra of the MLAT location of (a) northern and (b) southern EIA crest, the

TEC of (c) northern and (d) southern EIA crest, (e) EEJ and (f) Kp index during the same period as in

Figure 2. The white line in each panel represents the cone of influence of the wavelet analysis.

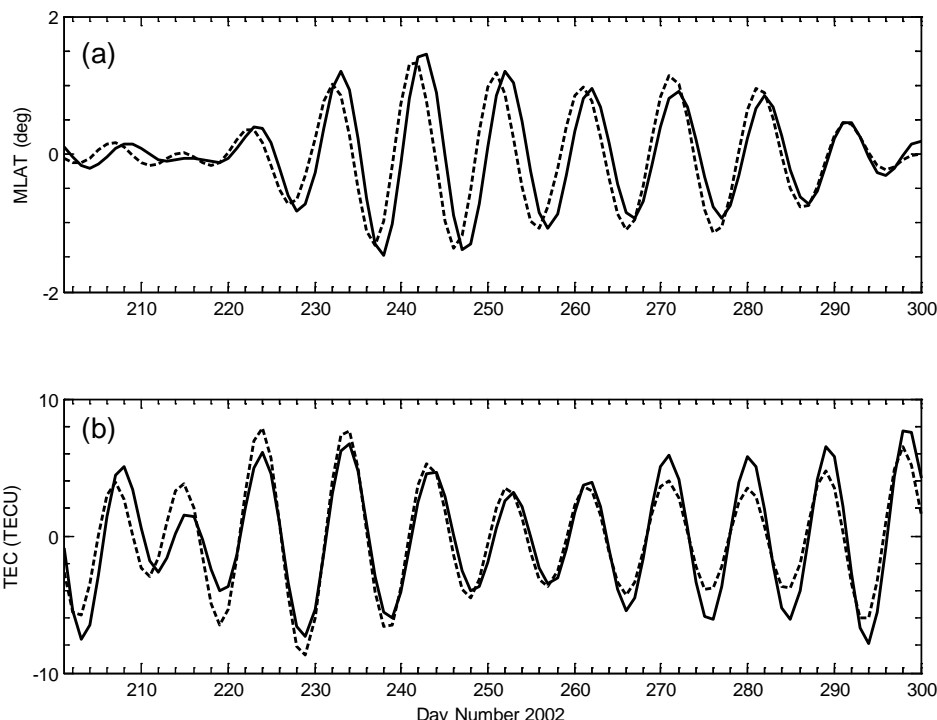


**Figure 6.** The band-pass filter results of the (a) MLAT location of (solid line) northern and (dash-dotted
line) southern EIA crest, the (b) TEC of (solid line) northern and (dash-dotted line) southern EIA crest

during the same period as in Figure 2.

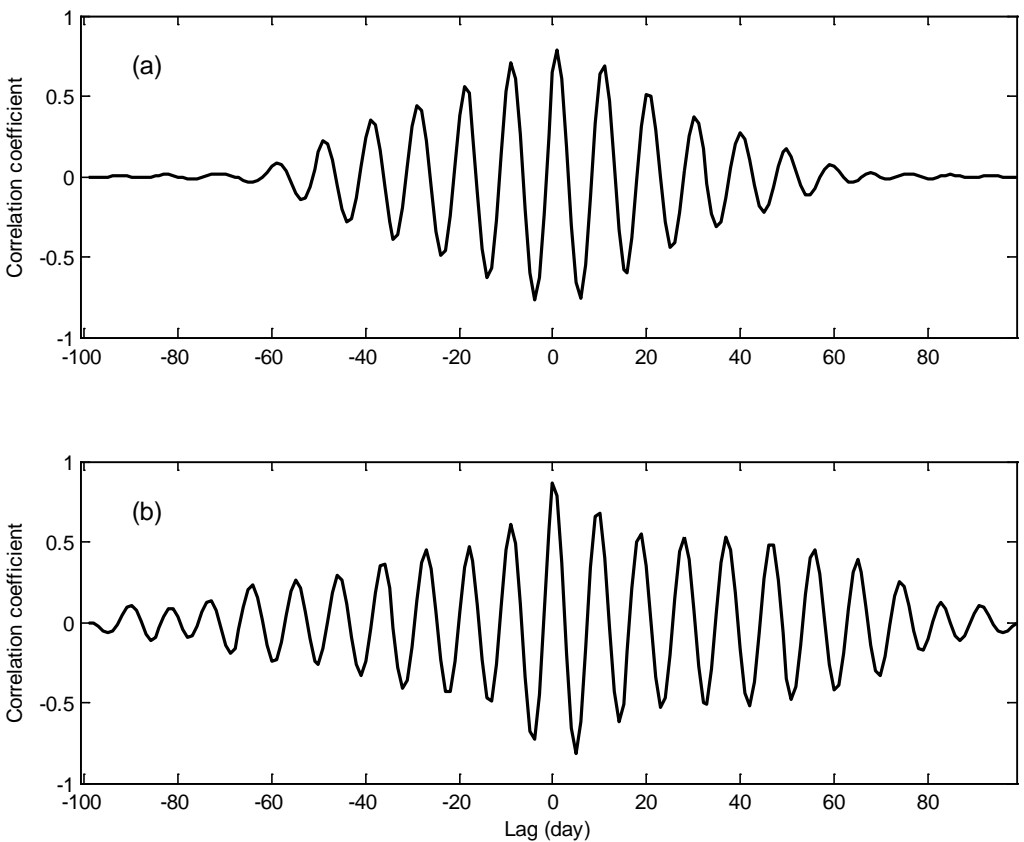


**Figure 7.** The cross-correlation of quasi 10-day waves in MLAT location (a) and TEC (b) between northern

and southern EIA crest.