# Peer review of "Quasi 10-day wave modulation of equatorial ionization anomaly during the Southern"

_Annales Geophysicae, 2019_

## Referee Comment (RC1) · Anonymous Referee #1 · 17 May 2019

In this work, the author has used the TEC data in the Southern Hemisphere (SH) to demonstrate the effects of quasi 10-day wave on the Northern and Southern TEC crests during the 2002 SH SSW event. The manuscript has been written well and is generally easy to understand. The results in this manuscript do provide a clear evidence of the quasi 10-day modulation of the TEC during the 2002 SSW event. I do a have a few concerns and comments, which are mentioned below. However, in general, the manuscript provides some interesting new results and should be accepted after a revision.

**Specific comments:**

Please plot a figure showing the location of the stations used in this work.

Line 97 - "To exclude these long period fluctuations in EIA region associated with solar/magnetosphere forcing, the periods longer than 15 days in the MLAT location and TEC of EIA crest are removed".
How is this process achieved? The author should clarify more regarding the applied method.

Line 152 - "Moreover, strong planetary wave scale quasi 10-day variation was observed in polar stratospheric temperature during this period".
Please cite the work in which this observation was mentioned.

**Technical corrections:**

Figure 1 caption - Correct to solar flux

Line 92 - it consists of an eastward-propagating

Line 105 - too weak to be identified in F10.7

Line 107 - evolution

Line 108 - TEC of EIA crest and Kp,

Line 124, 125 - band-pass

Line 146 - have

Line 155 - series of studies have showed

---

## Referee Comment (RC2) · Anonymous Referee #2 · 20 May 2019

Manuscript angeo-2019-43 Review on "Quasi 10-day wave modulation of equatorial ionization anomaly during the Southern Hemisphere stratospheric warming of 2002" by Xiaohua Mo

This paper is focused on the possible influence of the quasi 10-day planetary waves (PWs), registered in the high-latitude polar stratospheric temperature before and around the Southern Hemisphere (SH) sudden stratospheric warming (SSW) in 2002, on the oscillations of the equatorial ionization anomaly (EIA) crests and their Total Electron Content (TEC). The locations of the EIA crests are calculated from the observations of the two GPS stations in China which are situated near the northern and

southern EIA crests and the TEC data are derived from the International GNSS Service global ionospheric TEC maps in Asia. The SH SSW is described by the temperature and zonal mean zonal wind taken from the NCEP while the goemagnetic and solar variability are characterized by the Kp-index and solar radio flux F10.7 respectively. The period from July 21 to October 18, 2002 is considered (day numbers 200-300) and the quasi-10-day variability associated with the SSW is found in both the location of the EIA crests and the TEC between days 220-290. The author suggested that the observed ~10-day oscillation of the EIA region is generated through modulating the equatorial fountain effect.

The topic of the paper is certainly appropriate for the journal. In general, the paper is written clearly; actually it follows the pattern of the previous paper of the author, Mo et al. (2014) cited here. This study is certainly useful for the scientific community working on the vertical coupling of the atmosphere-ionosphere system however, however I think that it has serious deficiencies. Due to this I will suggest the publication of this paper but after serious revision and addressing the comments mentioned below.

Major comments:

(i) I have serious concern about the significance of the observed ~10-d oscillations particularly in the location of the EIA crests because the amplitude of these oscillations is only around $1.5°$ (Fig. 5a). Additionally, in Lomb-Scargle periodograms these oscillations are significant only above 90% confidence level (Fig. 3) that is not enough. It has been mentioned above that this study is similar to the previous one Mo et al. (2014) however, while there the quasi 16-day oscillations of the EIA crests were evident even in the raw data here the quasi-10-day ones are not. Usually only waves with significance at least above 95% confidence level are considered in styding the atmopsheric and ionospheric perturbations. The author does not mention anything about the error in calculating the coordinates of the EIA crests. Without knowing the error in calculating the MLAT of the EIA crests it is difficult to accept the 10-day variability of the EIA region as significant one.

(ii) In order to propose a mechanism for generating the 10-day variability of the EIA region the authors used indirect approach based on some general references on dynamics as well as references connected with the SH SSW in 2002. The important citations as: Eswaraiah et al. (2018) or Olson et al. (2013) which however present ground-based measurements at high latitudes or at Peruvian longitude sector cannot be considered as serious evidences because the author investigates different region, low latitudes over China. I cannot understand why the author does not use a meteor wind data from a Chinese radar at low latitudes and to check if there are quasi-10-day wave or modulated tides which are able to affect the fountain effect. Further, to see if the electric currents are modulated the author may consider the perturbations in the geomagnetic fields revealed from magnetometer measurements. Only then a solid evidence can be presented in support of the suggested mechnism.

(iii) Important studies on atmospheric dynamics and the ionospheric response to the SSW events are not cited.

Concrete comments:

P. 2, lines 37-38: Please, add the following references: Chau et al. (GRL, 36, L05101, 2009, doi:10.1029/2008GL036785) giving evidence for the vertical plasma drift changes during the SSW and Pancheva and Mukhtarov (JASTP, 73, 1697–1702, 2011, doi:10.1016/j.jastp.2011.03.006) presenting which main characteristics of the EIA and how they are changed during the major SSW.

P. 2, line 38: Please, add Jin et al. (JGR, 117, A10323, doi:10.1029/2012JA017650) where for the first time a comparison between the results from a whole atmosphere-ionosphere coupled model (GAIA) with the COSMIC and TIMED/SABER observations during the major SSW in January 2009 was presented.

P. 2, line 41: "Since planetary waves in the Southern Hemisphere (SH) generally have smaller amplitudes than in the Northern Hemisphere (NH)....." generally this is true only for the SPWs; the climatology of some other wellknown PWs, as for example, the

quasi-2-day W3 wave or the quasi-6-day W1 wave are both stronger in the SH.

P. 3, line 79: . . .. . .∼±15°N MLAT. . .; please, delete N

P. 4, line 117: "In additional, we . . ...."; please, delete al

P. 5, lines 126-127: "Note that quasi 10-day oscillations of northern and southern EIA crests are in-phase, which. . ..."; sorry, both oscillations are not in phase; the oscillation of the NH crest indicates a delay of a day with respect to the SH one. Please, calculate the cross-correlation function between both times, particularly between days 220-290 when they have large amplitudes, and will see that the largest cross-correlation will be found at different from zero time lag.

––––––––––––––––––––––

---

## Author Comment (AC1) · 9 Jun 2019

We very appreciate the referee's comments for our work and manuscript. Here are our reply comments. All major revisions are marked in yellow highlights.

In this work, the author has used the TEC data in the Southern Hemisphere (SH) to demonstrate the effects of quasi 10-day wave on the Northern and Southern TEC crests during the 2002 SH SSW event. The manuscript has been written well and is generally easy to understand. The results in this manuscript do provide a clear evidence of the quasi 10-day modulation of the TEC during the 2002 SSW event. I do a have a few concerns and comments, which are mentioned below. However, in general,

the manuscript provides some interesting new results and should be accepted after a revision.

Specific comments: 1. Please plot a figure showing the location of the stations used in this work. Answer: The locations of the GPS stations shown in Figures 1 are added in the revised version.

2. Line 97 - "To exclude these long period fluctuations in EIA region associated with solar/magnetosphere forcing, the periods longer than 15 days in the MLAT location and TEC of EIA crest are removed". How is this process achieved? The author should clarify more regarding the applied method. Answer: To remove the periods longer than 15 days in these parameters, these parameters are subtracted from their respective 15-day moving average.

3. Line 152 - "Moreover, strong planetary wave scale quasi 10-day variation was observed in polar stratospheric temperature during this period". Please cite the work in which this observation was mentioned. Answer: Relevant references have been cited in the revised version

Technical corrections: Figure 1 caption - Correct to solar flux Line 92 - it consists of an eastward-propagating Line 105 - too weak to be identified in F10.7 Line 107 - evolution Line 108 - TEC of EIA crest and Kp, Line 124, 125 - band-pass Line 146 - have Line 155 - series of studies have showed Answer: These grammatical and wording mistakes have been corrected in revised one.

Please also note the supplement to this comment:
https://www.ann-geophys-discuss.net/angeo-2019-43/angeo-2019-43-AC1-supplement.pdf

---

## Author Comment (AC2) · 9 Jun 2019

We very appreciate the referee's comments for our work and manuscript. Here are our reply comments. All major revisions are marked in yellow highlights.

Manuscript angeo-2019-43 Review on "Quasi 10-day wave modulation of equatorial ionization anomaly during the Southern Hemisphere stratospheric warming of 2002" by Xiaohua Mo

This paper is focused on the possible influence of the quasi 10-day planetary waves (PWs), registered in the high-latitude polar stratospheric temperature before and

around the Southern Hemisphere (SH) sudden stratospheric warming (SSW) in 2002, on the oscillations of the equatorial ionization anomaly (EIA) crests and their Total Electron Content (TEC). The locations of the EIA crests are calculated from the observations of the two GPS stations in China which are situated near the northern and southern EIA crests and the TEC data are derived from the International GNSS Service global ionospheric TEC maps in Asia. The SH SSW is described by the temperature and zonal mean zonal wind taken from the NCEP while the goemagnetic and solar variability are characterized by the Kp-index and solar radio flux F10.7 respectively. The period from July 21 to October 18, 2002 is considered (day numbers 200-300) and the quasi-10-day variability associated with the SSW is found in both the location of the EIA crests and the TEC between days 220-290. The author suggested that the observed $\sim$10-day oscillation of the EIA region is generated through modulating the equatorial fountain effect.

The topic of the paper is certainly appropriate for the journal. In general, the paper is written clearly; actually it follows the pattern of the previous paper of the author, Mo et al. (2014) cited here. This study is certainly useful for the scientific community working on the vertical coupling of the atmosphere-ionosphere system, however I think that it has serious deficiencies. Due to this I will suggest the publication of this paper but after serious revision and addressing the comments mentioned below.

Major comments: (i) I have serious concern about the significance of the observed $\sim$10-d oscillations particularly in the location of the EIA crests because the amplitude of these oscillations is only around 1.5$^\circ$ (Fig. 5a). Additionally, in Lomb-Scargle periodograms these oscillations are significant only above 90% confidence level (Fig. 3) that is not enough. It has been mentioned above that this study is similar to the previous one Mo et al. (2014). However, while there the quasi 16-day oscillations of the EIA crests were evident even in the raw data here the quasi-10-day ones are not. Usually only waves with significance at least above 95% confidence level are considered in studying the atmospheric and ionospheric perturbations. The author does not mention anything about the error in calculating the coordinates of the EIA crests. Without know-ing the error in calculating the MLAT of the EIA crests it is difficult to accept the 10-day variability of the EIA region as significant one. Answer: The perturbations amplitude in EIA region during SH SSW are smaller than those during NH SSW (Olson et al., 2013), so the Quasi 10-day wave in EIA region is not quite obvious. To make the results more credible, the 90% confidence level has been changed to 95% confidence level in Fig-ures 4. It can be seen the quasi 10-day periodic component of most EIA parameters can reach 95% confidence level. Moreover, the significance levels of quasi 10-day os-cillations revealed by the Morlet wavelet spectral analysis are higher than 95%. The error of the collected EIA crest latitude according to IPP trajectory is determined by the selected ionospheric shell height. Due to ionospheric variation, the shell height varies in different time and condition. Now, the constant shell height is used in almost all TEC derivation method, and the value of this height affects the resolution of the IPP location and the derived TEC values. So it is difficult to estimate the error of the EIA latitude. In the original manuscript, we only roughly estimate the error of the latitude from the sample rate of the raw GPS data according to the IPP velocity in the ionospheric shell.

(ii) In order to propose a mechanism for generating the 10-day variability of the EIA region the authors used indirect approach based on some general references on dy-namics as well as references connected with the SH SSW in 2002. The important citations as: Eswaraiah et al. (2018) or Olson et al. (2013) which however present ground-based measurements at high latitudes or at Peruvian longitude sector cannot be considered as serious evidences because the author investigates different region, low latitudes over China. I cannot understand why the author does not use a meteor wind data from a Chinese radar at low latitudes and to check if there are quasi-10-day wave or modulated tides which are able to affect the fountain effect. Further, to see if the electric currents are modulated the author may consider the perturbations in the geomagnetic fields revealed from magnetometer measurements. Only then a solid evidence can be presented in support of the suggested mechnism. Answer: The Meteor and MF radar was ever set up and was tested after 2003 in China, so the direct observations of MLT neutral wind are not available in China low latitude during 2002 SH SSW. The equatorial electrojet (EEJ) estimated by geomagnetic field cannot be obtained in China sector, because the geomagnetic field data in Bac Lieu in Vietnam are missing during the period from Aug 28, 2002 to Sep 26, 2002 (day number 240-269). In order to compensate this weakness, the EEJ estimated by geomagnetic field in Indian sector is added in revised manuscript. The EEJ in Indian sector also exhibit significant quasi 10-day periodic component.

(iii) Important studies on atmospheric dynamics and the ionospheric response to the SSW events are not cited. Concrete comments: P. 2, lines 37-38: Please, add the following references: Chau et al. (GRL, 36, L05101, 2009, doi:10.1029/2008GL036785) giving evidence for the vertical plasma drift changes during the SSW and Pancheva and Mukhtarov (JASTP, 73, 1697–1702, 2011, doi:10.1016/j.jastp.2011.03.006) presenting which main characteristics of the EIA and how they are changed during the major SSW. Answer: These references have been added in the introduction part of revised manuscript.

P. 2, line 38: Please, add Jin et al. (JGR, 117, A10323, doi:10.1029/2012JA017650) where for the first time a comparison between the results from a whole atmosphere ionosphere coupled model (GAIA) with the COSMIC and TIMED/SABER observations during the major SSW in January 2009 was presented. Answer: This reference has been added in the introduction part of revised manuscript.

P. 2, line 41: "Since planetary waves in the Southern Hemisphere (SH) generally have smaller amplitudes than in the Northern Hemisphere (NH)....." generally this is true only for the SPWs; the climatology of some other well known PWs, as for example, the quasi-2-day W3 wave or the quasi-6-day W1 wave are both stronger in the SH. Answer: Thank you for your remind. "planetary waves" has been changed to "stationary planetary waves".

P. 3, line 79:..... $\sim\pm15°$ N MLAT.... please, delete N Answer: Revised.

P. 4, line 117: "In additional, we . . . . . ."; please, delete al Answer: Revised.

P. 5, lines 126-127: "Note that quasi 10-day oscillations of northern and southern EIA crests are in-phase, which. . ."; sorry, both oscillations are not in phase; the oscillation of the NH crest indicates a delay of a day with respect to the SH one. Please, calculate the cross-correlation function between both times, particularly between days 220-290 when they have large amplitudes, and will see that the largest cross-correlation will be found at different from zero time lag. Answer: It is a very good comment. According to this suggestion, the cross-correlation function is used to reveal the phase relationship between northern and southern EIA crests. The results show that the quasi 10-day wave of northern EIA crest delay 1 day behind southern EIA crest.

Please also note the supplement to this comment:
https://www.ann-geophys-discuss.net/angeo-2019-43/angeo-2019-43-AC2-supplement.pdf

**Supplement:**

[revised manuscript text omitted]

---

## Author Response (AR1)

**Referee #1** Review on "Quasi 10-day wave modulation of equatorial ionization anomaly during the Southern Hemisphere stratospheric warming of 2002" by Xiaohua Mo

In this work, the author has used the TEC data in the Southern Hemisphere (SH) to demonstrate the effects of quasi 10-day wave on the Northern and Southern TEC crests during the 2002 SH SSW event. The manuscript has been written well and is generally easy to understand. The results in this manuscript do provide a clear evidence of the quasi 10-day modulation of the TEC during the 2002 SSW event. I do a have a few concerns and comments, which are mentioned below. However, in general, the manuscript provides some interesting new results and should be accepted after a revision.

**Answer:**

We very appreciate your substantial comments for our study about "Quasi 10-day wave modulation of equatorial ionization anomaly during the Southern Hemisphere stratospheric warming of 2002". Here are our reply comments. All major revisions are marked in yellow highlights. Thank you very much.

==============================================================

**Specific comments:**

……………………………………………………………………………………

1. Please plot a figure showing the location of the stations used in this work.

**Answer:** The locations of the GPS stations shown in Figures 1 are added in the revised version.

……………………………………………………………………………………

2. Line 97 - "To exclude these long period fluctuations in EIA region associated with solar/magnetosphere forcing, the periods longer than 15 days in the MLAT location and TEC of EIA crest are removed". How is this process achieved? The author should clarify more regarding the applied method.

**Answer:** To remove the periods longer than 15 days in these parameters, these parameters are subtracted from their respective 15-day moving average.

**Line # [101-104]:** To exclude these long period fluctuations in EIA region associated with solar/magnetosphere forcing, the periods longer than 15 days in the MLAT location and TEC of EIA crest, and EEJ are removed. Specifically, these parameters are subtracted from their respective 15-day moving average.

……………………………………………………………………………………

3. Line 152 - "Moreover, strong planetary wave scale quasi 10-day variation was observed in polar stratospheric temperature during this period". Please cite the work in which this observation was mentioned.

**Answer:** Relevant references have been cited in the revised version.

**Line # [162-164]:** Moreover, strong planetary wave scale quasi 10-day variation was observed in polar stratospheric temperature during this period (Krüger et al., 2005; Palo et al., 2005), so the quasi 10-day oscillations in EIA region may be related to atmosphere perturbations linking the SSW in the Southern

Hemisphere.

…………………………………………………………………………………

**Technical corrections:**

Figure 1 caption - Correct to solar flux

Line 92 - it consists of an eastward-propagating

Line 105 - too weak to be identified in F10.7

Line 107 - evolution

Line 108 - TEC of EIA crest and Kp,

Line 124, 125 - band-pass

Line 146 - have

Line 155 - series of studies have showed

**Answer:** These grammatical and wording mistakes have been corrected in revised one.

…………………………………………………………………………………

+++++++++++++++++++++++++++++++++++++++++++++++++++++++++++++++

**Referee #2** Review on "Quasi 10-day wave modulation of equatorial ionization anomaly during the Southern Hemisphere stratospheric warming of 2002" by Xiaohua Mo

This paper is focused on the possible influence of the quasi 10-day planetary waves (PWs), registered in the high-latitude polar stratospheric temperature before and around the Southern Hemisphere (SH) sudden stratospheric warming (SSW) in 2002, on the oscillations of the equatorial ionization anomaly (EIA) crests and their Total Electron Content (TEC). The locations of the EIA crests are calculated from the observations of the two GPS stations in China which are situated near the northern and southern EIA crests and the TEC data are derived from the International GNSS Service global ionospheric TEC maps in Asia. The SH SSW is described by the temperature and zonal mean zonal wind taken from the NCEP while the goemagnetic and solar variability are characterized by the Kp-index and solar radio flux F10.7 respectively. The period from July 21 to October 18, 2002 is considered (day numbers 200-300) and the quasi-10-day variability associated with the SSW is found in both the location of the EIA crests and the TEC between days 220-290. The author suggested that the observed ~10-day oscillation of the EIA region is generated through modulating the equatorial fountain effect.

The topic of the paper is certainly appropriate for the journal. In general, the paper is written clearly; actually it follows the pattern of the previous paper of the author, Mo et al. (2014) cited here. This study is certainly useful for the scientific community working on the vertical coupling of the atmosphere-ionosphere system, however I think that it has serious deficiencies. Due to this I will suggest the publication of this paper but after serious revision and addressing the comments mentioned below.

**Answer:**

We very appreciate your substantial comments for our study about "Quasi 10-day wave modulation of equatorial ionization anomaly during the Southern Hemisphere stratospheric warming of 2002". Here are our reply comments. All major revisions are marked in yellow highlights. Thank you very much.

============================================================

**Major comments:**

……………………………………………………………………………………

(1) I have serious concern about the significance of the observed ~10-d oscillations particularly in the location of the EIA crests because the amplitude of these oscillations is only around 1.5° (Fig. 5a). Additionally, in Lomb-Scargle periodograms these oscillations are significant only above 90% confidence level (Fig. 3) that is not enough. It has been mentioned above that this study is similar to the previous one Mo et al. (2014). However, while there the quasi 16-day oscillations of the EIA crests were evident even in the raw data here the quasi-10-day ones are not. Usually only waves with significance at least above 95% confidence level are considered in studying the atmospheric and ionospheric perturbations. The author does not mention anything about the error in calculating the coordinates of the EIA crests. Without knowing the error in calculating the MLAT of the EIA crests it is difficult to accept the 10-day variability of the EIA region as significant one.

**Answer:** The perturbations amplitude in EIA region during SH SSW are smaller than those during NH SSW (Olson et al., 2013), so the Quasi 10-day wave in EIA region is not quite obvious. To make the results more credible, the 90% confidence level has been changed to 95% confidence level in Figures 4. It can be seen the quasi 10-day periodic component of most EIA parameters can reach 95% confidence level. Moreover, the significance levels of quasi 10-day oscillations revealed by the Morlet wavelet spectral analysis are higher than 95%. The error of the collected EIA crest latitude according to IPP trajectory is determined by the selected ionospheric shell height. Due to ionospheric variation, the shell height varies in different time and condition. Now, the constant shell height is used in almost all TEC derivation method, and the value of this height affects the resolution of the IPP location and the derived TEC values. So it is difficult to estimate the error of the EIA latitude. In the original manuscript, we only roughly estimate the error of the latitude from the sample rate of the raw GPS data according to the IPP velocity in the ionospheric shell.

**Line #[104-108]:** The residuals are subjected to Lomb-Scargle (L-S) spectral analysis (Lomb,1976; Scargle, 1982), and the results are shown in Figures 4a, 4b, 4c, 4d, and 4e. The horizontal dashed lines represent the 95% confidence level. It is evident that the MLAT location and TEC of EIA crest, and EEJ all exhibit significant quasi 10-day periodic component, which exceed or approach 95% confidence level, suggesting that the whole dynamical process in EIA region is modulated by quasi 10-day wave.

**Line # [69-74]:** Since the ionospheric vertical TEC usually reach the maximum at EIA crest, the location of EIA crest can be obtained by vertical TEC values at each ionospheric penetration point (IPP), which is the intersection of the line of sight and the ionospheric shell (assumed to be 400 km) (Mo et al., 2014). The relative accuracy of the TEC is 0.02 total electron content unit ($1TECU=10^{16}$ el m$^{-2}$) (Hofmann-Wellenhof et al., 1992). The sample rate of these GPS stations were 30s, so the resolution of the location of EIA crest is less than 25 km (Mo et al., 2017).

……………………………………………………………………………………

(2) In order to propose a mechanism for generating the 10-day variability of the EIA region the authors used indirect approach based on some general references on dynamics as well as references connected with the SH SSW in 2002. The important citations as: Eswaraiah et al. (2018) or Olson et al. (2013) which however present ground-based measurements at high latitudes or at Peruvian longitude sector cannot be considered as serious evidences because the author investigates different region, low latitudes over China. I cannot understand why the author does not use a meteor wind data from a Chinese radar at low latitudes and to check if there are quasi-10-day wave or modulated tides which are able to affect the fountain effect. Further, to see if the electric currents are modulated the author may consider the perturbations in the geomagnetic fields revealed from magnetometer measurements. Only then a solid evidence can be presented in support of the suggested mechnism.

**Answer:** The Meteor and MF radar was ever set up and was tested after 2003 in China, so the direct observations of MLT neutral wind are not available in China low latitude during 2002 SH SSW. The equatorial electrojet (EEJ) estimated by geomagnetic field cannot be obtained in China sector, because the geomagnetic field data in Bac Lieu in Vietnam are missing during the period from Aug 28, 2002 to Sep 26, 2002 (day number 240-269). In order to compensate this weakness, the EEJ estimated by geomagnetic field in Indian sector is added in revised manuscript. The EEJ in Indian sector also exhibit significant quasi 10-day periodic component.

**Line # [83-86]:** To demonstrate the dynamical process in EIA region, the EEJ is also used in this study, which can be estimated by the difference between the horizontal component of geomagnetic field at TIR (8.7°N, 77.8°E, MLAT~0.03°N) and VSK (17.68°N, 83.32°E, MLAT~8.56°N) (Rastogi et al., 1990). The results are shown in Figures 2e.

……………………………………………………………………………………

(3) Important studies on atmospheric dynamics and the ionospheric response to the SSW events are not cited.

Concrete comments:

P. 2, lines 37-38: Please, add the following references: Chau et al. (GRL, 36, L05101, 2009, doi:10.1029/2008GL036785) giving evidence for the vertical plasma drift changes during the SSW and Pancheva and Mukhtarov (JASTP, 73, 1697–1702, 2011, doi:10.1016/j.jastp.2011.03.006) presenting which main characteristics of the EIA and how they are changed during the major SSW.

P. 2, line 38: Please, add Jin et al. (JGR, 117, A10323, doi:10.1029/2012JA017650) where for the first time a comparison between the results from a whole atmosphere ionosphere coupled model (GAIA) with the COSMIC and TIMED/SABER observations during the major SSW in January 2009 was presented.

**Answer:** These references have been added in the introduction part of revised manuscript.

**Line # [35-38]:** Although the main processes of SSW occur in the middle atmosphere, its effects on the ionosphere have been observed in significant changes of equatorial electrojet (EEJ), vertical plasma drift, and equatorial ionization anomaly (EIA) (Vineeth et al., 2007; Chau et al., 2009; Goncharenko et al., 2010; Pancheva and Mukhtarov, 2011; Jin et al., 2012).

……………………………………………………………………………………

P. 2, line 41: "Since planetary waves in the Southern Hemisphere (SH) generally have smaller amplitudes than in the Northern Hemisphere (NH)….." generally this is true only for the SPWs; the climatology of some other well known PWs, as for example, the quasi-2-day W3 wave or the quasi-6-day W1 wave are both stronger in the SH.

**Answer:** Thank you for your remind. "planetary waves" has been changed to "stationary planetary waves".

……………………………………………………………………………………………

P. 3, line 79:….. ~±15º N MLAT…. please, delete N

**Answer:** Revised.

……………………………………………………………………………………………

P. 4, line 117: "In additional, we ……"; please, delete al

**Answer:** Revised.

……………………………………………………………………………………………

P. 5, lines 126-127: "Note that quasi 10-day oscillations of northern and southern EIA crests are in-phase, which…"; sorry, both oscillations are not in phase; the oscillation of the NH crest indicates a delay of a day with respect to the SH one. Please, calculate the cross-correlation function between both times, particularly between days 220-290 when they have large amplitudes, and will see that the largest cross-correlation will be found at different from zero time lag.

**Answer:** It is a very good comment. According to this suggestion, the cross-correlation function is used to reveal the 
[revised manuscript text omitted]

---

## Referee Report (RR1)

**Review of ANGEO Manuscript**

**Quasi 10-day wave modulation of equatorial ionization anomaly during the Southern** 1
**Hemisphere stratospheric warming of 2002**

Xiaohua Mo and Donghe Zhang

Using the location and TEC of EIA crests derived from GPS station observations and GIMs, the authors report on a quasi 10-day periodic variability in northern and southern EIA region in Asian sector during the 2002 SH SSW. Around the same time period, this quasi 10-day oscillation is also seen in the polar stratospheric temperature and EEJ, which is absent and weak in Kp and F10.7 index, respectively. Given previous work showing a strong quasi 10-day planetary wave with zonal wave numbers s=1 extend from the lower stratosphere to mesosphere and lower thermosphere the authors infer that the quasi 10-day variation in the EIA region should be ascribed to enhanced 10-day planetary wave in lower atmosphere associated with SSW.

**General comment:**

This manuscript contains some interesting results and I find it improved in this revised version. In particular I find the results of the 10-day modulation of the TEC during the 2002 SH SWW event compelling. A major source of concern is lack of sufficient evidence on potential connections between the 10-day periodic variation of EIA crests and SSW/EEJ. Further statistical analysis that demonstrates this connection is required before publication can be recommended.

**Other comments**

- The abstract is not sufficiently developed.
- Line 23 'Norther Hemisphere (SH)', is this a typo or the authors mean both NH and SH?
- Line 54: 'The researches', change to 'Research'.
- Unit missing in the legend of Figure 5.
- Line 166: 'A series of studies', which ones? Include appropriate references.
- Line 208-221: While the observed 10-day oscillation may be ascribable to the SSW the manuscript presents insufficient evidence to justify this claim.

---

## Author Response (AR2)

Review on "Quasi 10-day wave modulation of equatorial ionization anomaly during the Southern Hemisphere stratospheric warming of 2002" by Xiaohua Mo and Donghe Zhang

Using the location and TEC of EIA crests derived from GPS station observations and GIMs, the authors report on a quasi 10-day periodic variability in northern and southern EIA region in Asian sector during the 2002 SH SSW. Around the same time period, this quasi 10-day oscillation is also seen in the polar stratospheric temperature and EEJ, which is absent and weak in Kp and F10.7 index, respectively. Given previous work showing a strong quasi 10-day planetary wave with zonal wave numbers s=1 extend from the lower stratosphere to mesosphere and lower thermosphere the authors infer that the quasi 10-day variation in the EIA region should be ascribed to enhanced 10-day planetary wave in lower atmosphere associated with SSW.

**General comment:**

This manuscript contains some interesting results and I find it improved in this revised version. In particular I find the results of the 10-day modulation of the TEC during the 2002 SH SWW event compelling. A major source of concern is lack of sufficient evidence on potential connections between the 10-day periodic variation of EIA crests and SSW/EEJ. Further statistical analysis that demonstrates this connection is required before publication can be recommended.

**Answer:**

We very appreciate your substantial comments for our study about "Quasi 10-day wave modulation of equatorial ionization anomaly during the Southern Hemisphere stratospheric warming of 2002". The connections between the 10-day periodic variation of EIA crests and SSW/EEJ have been further analyzed and discussed in the introduction and discussions part of revised manuscript. All major revisions are marked in yellow highlights. Thank you very much.

[revised manuscript text omitted]

………………………………………………………………………………………

2. Line 23 'Norther Hemisphere (SH)', is this a typo or the authors mean both NH and SH?

**Answer:** This is a typo, and the 'Northern Hemisphere (SH)' has been changed to 'Northern Hemisphere (NH)'.

………………………………………………………………………………………

3. Line 54: 'The researches', change to 'Research'.

**Answer:** the 'researches' has been changed to 'research'.

………………………………………………………………………………………

4. Unit missing in the legend of Figure 5.

**Answer:** The color bar number in Figure 5 is the wavelet power with no units.

………………………………………………………………………………………

5. Line 166: 'A series of studies', which ones? Include appropriate references.

**Answer:** the corresponding references have been cited in the revised version.

Line # [192-194]: A series of studies have showed how the quasi 10-day planetary wave in stratosphere can penetrate into the ionosphere E region (Krüger et al., 2005; Palo et al., 2005; Chang et al., 2009).

………………………………………………………………………………………

6. Line 208-221: While the observed 10-day oscillation may be ascribable to the SSW the manuscript presents insufficient evidence to justify this claim.

**Answer:** The connections between the 10-day periodic variation of EIA crests and SSW have been further analyzed and discussed in the introduction and discussions parts of revised manuscript. In the conclusions part, we summarize these results as follows:

[revised manuscript text omitted]